# SLICED KERNELIZED STEIN DISCREPANCY

**Wenbo Gong**
University of Cambridge
wg242@cam.ac.uk

**Yingzhen Li** [*]
Imperial College London
yingzhen.li@imperial.ac.uk

**José Miguel Hernández-Lobato**
University of Cambridge
The Alan Turing Institute
jmh233@cam.ac.uk

## ABSTRACT

Kernelized Stein discrepancy (KSD), though being extensively used in goodness-of-fit tests and model learning, suffers from the curse-of-dimensionality. We address this issue by proposing the *sliced Stein discrepancy* and its scalable and kernelized variants, which employ kernel-based test functions defined on the optimal one-dimensional projections. When applied to goodness-of-fit tests, extensive experiments show the proposed discrepancy significantly outperforms KSD and various baselines in high dimensions. For model learning, we show its advantages over existing Stein discrepancy baselines by training independent component analysis models with different discrepancies. We further propose a novel particle inference method called *sliced Stein variational gradient descent* (S-SVGD) which alleviates the mode-collapse issue of SVGD in training variational autoencoders.

## 1 INTRODUCTION

Discrepancy measures for quantifying differences between two probability distributions play key roles in statistics and machine learning. Among many existing discrepancy measures, Stein discrepancy (SD) is unique in that it only requires samples from one distribution and the score function (i.e. the gradient up to a multiplicative constant) from the other (Gorham & Mackey, 2015). SD, a special case of *integral probability metric* (IPM) (Sriperumbudur et al., 2009), requires finding an optimal test function within a given function family. This optimum is analytic when a *reproducing kernel Hilbert space* (RKHS) is used as the test function family, and the corresponding SD is named *kernelized Stein discrepancy* (KSD) (Liu et al., 2016; Chwialkowski et al., 2016). Variants of SDs have been widely used in both Goodness-of-fit (GOF) tests (Liu et al., 2016; Chwialkowski et al., 2016) and model learning (Liu & Feng, 2016; Grathwohl et al., 2020; Hu et al., 2018; Liu & Wang, 2016).

Although theoretically elegant, KSD, especially with RBF kernel, suffers from the "curse-of-dimensionality" issue, which leads to significant deterioration of test power in GOF tests (Chwialkowski et al., 2016; Huggins & Mackey, 2018) and mode collapse in particle inference (Zhuo et al., 2017; Wang et al., 2018). A few attempts have been made to address this problem, however, they either are limited to specific applications with strong assumptions (Zhuo et al., 2017; Chen & Ghattas, 2020; Wang et al., 2018) or require significant approximations (Singhal et al., 2019). As an alternative, in this work we present our solution to this issue by adopting the idea of "slicing". Here the key idea is to project the score function and test inputs onto multiple one dimensional slicing directions, resulting in a variant of SD that only requires to work with one-dimensional inputs for the test functions. Specifically, our contributions are as follows.

- We propose a novel theoretically validated family of discrepancies called *sliced Stein discrepancy* (SSD), along with its scalable variant called *max sliced kernelized Stein discrepancy* (maxSKSD) using kernel tricks and the *optimal test directions*.
- A GOF test is derived based on an unbiased estimator of maxSKSD with optimal test directions. MaxSKSD achieves superior performance on benchmark problems and *restricted Boltzmann machine* models (Liu et al., 2016; Huggins & Mackey, 2018).

---

[*]Work done at Microsoft Research Cambridge

- We evaluate the maxSKSD in model learning by two schemes. First, we train an independent component analysis (ICA) model in high dimensions by directly minimising maxSKSD, which results in faster convergence compared to baselines (Grathwohl et al., 2020). Further, we propose a particle inference algorithm based on maxSKSD called the *sliced Stein variational gradient descent* (S-SVGD) as a novel variant of the original SVGD (Liu & Wang, 2016). It alleviates the posterior collapse of SVGD when applied to training variational autoencoders (Kingma & Welling, 2013; Rezende et al., 2014).

## 2 BACKGROUND

### 2.1 KERNELIZED STEIN DISCREPANCY

For two probability distributions $p$ and $q$ supported on $\mathcal{X} \subseteq \mathbb{R}^D$ with continuous differentiable densities $p(\boldsymbol{x})$ and $q(\boldsymbol{x})$, we define the score $\boldsymbol{s}_p(\boldsymbol{x}) = \nabla_{\boldsymbol{x}} \log p(\boldsymbol{x})$ and $\boldsymbol{s}_q(\boldsymbol{x})$ accordingly. For a test function $f : \mathcal{X} \to \mathbb{R}^D$, the Stein operator is defined as

$$\mathcal{A}_p f(\boldsymbol{x}) = \boldsymbol{s}_p(\boldsymbol{x})^T f(\boldsymbol{x}) + \nabla_{\boldsymbol{x}}^T f(\boldsymbol{x}). \tag{1}$$

For a function $f_0 : \mathbb{R}^D \to \mathbb{R}$, the *Stein class* $\mathcal{F}_q$ of $q$ is defined as the set of functions satisfying Stein's identity (Stein et al., 1972): $\mathbb{E}_q[\boldsymbol{s}_q(\boldsymbol{x})f_0(\boldsymbol{x}) + \nabla_{\boldsymbol{x}}f_0(\boldsymbol{x})] = \boldsymbol{0}$. This can be generalized to a vector function $\boldsymbol{f} : \mathbb{R}^D \to \mathbb{R}^D$ where $\boldsymbol{f} = [f_1(\boldsymbol{x}), \ldots, f_D(\boldsymbol{x})]^T$ by letting $f_i$ belongs to the Stein class of $q$ for each $i \in D$. Then the Stein discrepancy (Liu et al., 2016; Gorham & Mackey, 2015) is defined as

$$D(q, p) = \sup_{f \in \mathcal{F}_q} \mathbb{E}_q[\mathcal{A}_p f(\boldsymbol{x})] = \sup_{f \in \mathcal{F}_q} \mathbb{E}_q[(\boldsymbol{s}_p(\boldsymbol{x}) - \boldsymbol{s}_q(\boldsymbol{x}))^T f(\boldsymbol{x})]. \tag{2}$$

When $\mathcal{F}_q$ is sufficiently rich, and $q$ vanishes at the boundary of $\mathcal{X}$, the supremum is obtained at $f^*(\boldsymbol{x}) \propto \boldsymbol{s}_p(\boldsymbol{x}) - \boldsymbol{s}_q(\boldsymbol{x})$ with some mild regularity conditions on $f$ (Hu et al., 2018). Thus, the Stein discrepancy focuses on the score difference of $p$ and $q$. *Kernelized Stein discrepancy* (KSD) (Liu et al., 2016; Chwialkowski et al., 2016) restricts the test functions to be in a $D$-dimensional RKHS $\mathcal{H}_D$ with kernel $k$ to obtain an analytic form. By defining $u_p(\boldsymbol{x}, \boldsymbol{x}') = \boldsymbol{s}_p(\boldsymbol{x})^T \boldsymbol{s}_p(\boldsymbol{x}')k(\boldsymbol{x}, \boldsymbol{x}') + \boldsymbol{s}_p(\boldsymbol{x})^T \nabla_{\boldsymbol{x}'}k(\boldsymbol{x}, \boldsymbol{x}') + \boldsymbol{s}_p(\boldsymbol{x}')^T \nabla_{\boldsymbol{x}}k(\boldsymbol{x}, \boldsymbol{x}') + \text{Tr}(\nabla_{\boldsymbol{x}, \boldsymbol{x}'}k(\boldsymbol{x}, \boldsymbol{x}'))$ the analytic form of KSD is:

$$D^2(q, p) = \left( \sup_{f \in \mathcal{H}_D, ||f||_{\mathcal{H}_D} \leq 1} \mathbb{E}_q[\mathcal{A}_p f(\boldsymbol{x})] \right)^2 = \mathbb{E}_{q(\boldsymbol{x})q(\boldsymbol{x}')}[u_p(\boldsymbol{x}, \boldsymbol{x}')]. \tag{3}$$

### 2.2 STEIN VARIATIONAL GRADIENT DESCENT

Although SD and KSD can be directly minimized for variational inference (VI) (Ranganath et al., 2016; Liu & Feng, 2016; Feng et al., 2017), Liu & Wang (2016) alternatively proposed a novel particle inference algorithm called *Stein variational gradient descent* (SVGD). It applies a sequence of deterministic transformations to a set of points such that each of mappings maximally decreases the Kullback-Leibler (KL) divergence from the particles' underlying distribution $q$ to the target $p$.

To be specific, we define the mapping $T(\boldsymbol{x}) : \mathbb{R}^D \to \mathbb{R}^D$ as $T(\boldsymbol{x}) = \boldsymbol{x} + \epsilon\phi(\boldsymbol{x})$ where $\phi$ characterises the perturbations. The result from Liu & Wang (2016) shows that the optimal perturbation inside the RKHS is exactly the optimal test function in KSD.

**Lemma 1.** *(Liu & Wang, 2016) Let $T(\boldsymbol{x}) = \boldsymbol{x} + \epsilon\phi(\boldsymbol{x})$ and $q_{[T]}(\boldsymbol{z})$ be the density of $\boldsymbol{z} = T(\boldsymbol{x})$ when $\boldsymbol{x} \sim q(\boldsymbol{x})$. If the perturbation $\phi$ is in the RKHS $\mathcal{H}_D$ and $||\phi||_{\mathcal{H}_D} \leq D(q, p)$, then the steepest descent directions $\phi_{q,p}^*$ is*

$$\phi_{q,p}^*(\cdot) = \mathbb{E}_q[\nabla_{\boldsymbol{x}} \log p(\boldsymbol{x})k(\boldsymbol{x}, \cdot) + \nabla_{\boldsymbol{x}}k(\boldsymbol{x}, \cdot)] \tag{4}$$

*and $\nabla_\epsilon KL[q_{[T]}||p]|_{\epsilon=0} = -D^2(q, p)$.*

The first term in Eq.(4) is called drift, which drives the particles towards a mode of $p$. The second term controls the repulsive force, which spreads the particles around the mode. When particles stop moving, the KL decrease magnitude $\epsilon D^2(q, p)$ is 0, which means the KSD is zero and $p = q$ a.e.

## 3 SLICED KERNELIZED STEIN DISCREPANCY

We propose the *sliced Stein discrepancy* (SSD) and kernelized version named maxSKSD. Theoretically, we prove their correctness as discrepancy measures. Methodology-wise, we apply maxSKSD to GOF tests, and develop two ways for model learning.

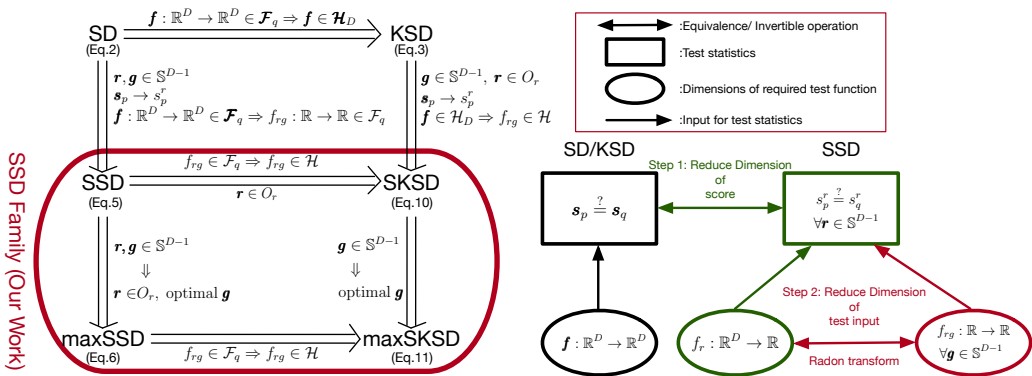

Figure 1: **(Left)** The connections between SD, KSD and the proposed SSD family. **(Right)** The intuition of SSD. The rectangular boxes indicate what statistics the discrepancy wants to test. The circle represents the dimension of the test function required for such test. The double arrow means equivalence relations or invertable operations.

## 3.1 SLICED STEIN DISCREPANCY

Before moving to the details, we give a brief overview of the intuition on how to tackle the curse-of-fimensionality issue of SD (The right figure of Figure 1). For detailed explanation, refer to appendix B.1. This issue of Stein discrepancy (Eq.2) comes from two sources: the score function $s_p(\boldsymbol{x})$ and the test function $f(\boldsymbol{x})$ defined on $\mathcal{X} \subset \mathbb{R}^D$. First, we notice that comparing $\boldsymbol{s}_p$ and $\boldsymbol{s}_q$ is equivalent to comparing projected score $s_p^r = \boldsymbol{s}_p^T \boldsymbol{r}$ and $s_q^r$ for all $\boldsymbol{r} \in \mathbb{S}^{D-1}$ on an hyper-sphere (Green square in Figure 1 (Right)). This operation reduces the test function's output from $\mathbb{R}^D$ to $\mathbb{R}$ (Green circle in Figure 1 (Right)). However, its input dimension is not affected. Reducing the input dimension of test functions is non-trivial, as directly removing input dimensions results in the test power decrease. This is because less information is accessed by the test function (see examples in appendix B.1). Our solution to this problem uses *Radon transform* which is inspired by CT-scans. It projects the original test function $f(\boldsymbol{x})$ in Stein discrepancy (Eq. 2) (as an $\mathbb{R}^D \to \mathbb{R}$ mapping) to a group of $\mathbb{R} \to \mathbb{R}$ functions along a set of directions ($\boldsymbol{g} \in \mathbb{S}^{D-1}$). Then, this group of functions are used as the new test functions to define the proposed discrepancy. The invertibility of *Radon transform* ensures that testing with input in the original space $\mathbb{R}^D$ is equivalent to the test using a group of low dimensional functions with input in $\mathbb{R}$. Thus, the above two steps not only reduce the dimensions of the test function's output and input, but also maintain the validity of the resulting discrepancy as each step is either equivalent or invertible.

In detail, assume two distributions $p$ and $q$ supported on $\mathbb{R}^D$ with differentiable densities $p(\boldsymbol{x})$ and $q(\boldsymbol{x})$, and define the test functions $f(\cdot; \boldsymbol{r}, \boldsymbol{g}) : \mathbb{R}^D \to \mathbb{R}$ such that $f(\boldsymbol{x}; \boldsymbol{r}, \boldsymbol{g}) = f_{rg} \circ h_g(\boldsymbol{x}) = f_{rg}(\boldsymbol{x}^T \boldsymbol{g})$, where $h_g(\cdot)$ is the inner product with $\boldsymbol{g}$ and $f_{rg} : \mathbb{R} \to \mathbb{R}$. One should note that the $\boldsymbol{r}$ and $\boldsymbol{g}$ in $f(\cdot; \boldsymbol{r}, \boldsymbol{g})$ should not just be treated as parameters in a test function $f$. In fact, they are more like the index to indicate that for each pair of $\boldsymbol{r}, \boldsymbol{g}$, we need a new $f(\cdot; \boldsymbol{r}, \boldsymbol{g})$, i.e. new $f_{rg}$, which is completely independent to other test functions. The proposed sliced Stein discrepancy (SSD), defined using two uniform distributions $p_r(\boldsymbol{r})$ and $p_g(\boldsymbol{g})$ over the hypersphere $\mathbb{S}^{D-1}$, is given by the following, with $f_{rg} \in \mathcal{F}_q$ meaning $f(\cdot; \boldsymbol{r}, \boldsymbol{g}) \in \mathcal{F}_q$:

$$S(q, p) = \mathbb{E}_{p_r, p_g} \left[ \sup_{f_{rg} \in \mathcal{F}_q} \mathbb{E}_q[s_p^r(\boldsymbol{x}) f_{rg}(\boldsymbol{x}^T \boldsymbol{g}) + \boldsymbol{r}^T \boldsymbol{g} \nabla_{\boldsymbol{x}^T \boldsymbol{g}} f_{rg}(\boldsymbol{x}^T \boldsymbol{g})] \right]. \quad (5)$$

We verify the proposed SSD is a valid discrepancy measure, namely, $S(q, p) = 0$ iff. $q = p$ a.e.

**Theorem 1.** *(SSD Validity) If assumptions 1-4 in appendix A are satisfied, then for two probability distributions $p$ and $q$, $S(q, p) \geq 0$, and $S(q, p) = 0$ if and only if $p = q$ a.e.*

Despite this attractive theoretical result, SSD is difficult to compute in practice. Specifically, the expectations over $\boldsymbol{r}$ and $\boldsymbol{g}$ can be approximated by Monte Carlo but this typically requires a very

large number of samples in high dimensions (Deshpande et al., 2019). We propose to relax such limitations by using only a finite number of slicing directions $\boldsymbol{r}$ from an orthogonal basis $O_r$ of $\mathbb{R}^D$, e.g. the standard basis of one-hot vectors, and the corresponding optimal test direction $\boldsymbol{g}_r$ for each $\boldsymbol{r}$. We call this variant maxSSD, which is defined as follows and validated in Corollary 1.1:

$$S_{max}(q,p) = \sum_{\boldsymbol{r} \in O_r} \sup_{f_{rg_r} \in \mathcal{F}_q, \boldsymbol{g}_r \in \mathbb{S}^{D-1}} \mathbb{E}_q[s_p^r(\boldsymbol{x})f_{rg_r}(\boldsymbol{x}^T\boldsymbol{g}_r) + \boldsymbol{r}^T\boldsymbol{g}_r\nabla_{\boldsymbol{x}^T\boldsymbol{g}_r}f_{rg_r}(\boldsymbol{x}^T\boldsymbol{g}_r)]. \quad (6)$$

**Corollary 1.1.** *(maxSSD) Assume the conditions in Theorem 1, then $S_{max}(q,p) = 0$ iff. $p = q$ a.e.*

## 3.2 CLOSED FORM SOLUTION WITH THE KERNEL TRICK

The optimal test function given $\boldsymbol{r}$ and $\boldsymbol{g}$ is intractable without further assumptions on the test function families. This introduces another scalability issue as optimizing these test functions explicitly can be time consuming. Fortunately, we can apply the kernel trick to obtain its analytic form. Assume for each test function $f_{rg} \in \mathcal{H}_{rg}$, where $\mathcal{H}_{rg}$ is a scalar-valued RKHS equipped with kernel $k(\boldsymbol{x}, \boldsymbol{x}'; \boldsymbol{r}, \boldsymbol{g}) = k_{rg}(\boldsymbol{x}^T\boldsymbol{g}, \boldsymbol{x}'^T\boldsymbol{g})$ that satisfies assumption 5 in appendix A and $f_{rg}(\boldsymbol{x}^T\boldsymbol{g}) = \langle f_{rg}, k_{rg}(\boldsymbol{x}^T\boldsymbol{g}, \cdot)\rangle_{\mathcal{H}_{rg}}$. We define the following quantities:

$$\xi_{p,r,g}(\boldsymbol{x}, \cdot) = s_p^r(\boldsymbol{x})k_{rg}(\boldsymbol{x}^T\boldsymbol{g}, \cdot) + \boldsymbol{r}^T\boldsymbol{g}\nabla_{\boldsymbol{x}^T\boldsymbol{g}}k_{rg}(\boldsymbol{x}^T\boldsymbol{g}, \cdot), \quad (7)$$

$$h_{p,r,g}(\boldsymbol{x}, \boldsymbol{y}) = s_p^r(\boldsymbol{x})k_{rg}(\boldsymbol{x}^T\boldsymbol{g}, \boldsymbol{y}^T\boldsymbol{g})s_p^r(\boldsymbol{y}) + \boldsymbol{r}^T\boldsymbol{g}s_p^r(\boldsymbol{y})\nabla_{\boldsymbol{x}^T\boldsymbol{g}}k_{rg}(\boldsymbol{x}^T\boldsymbol{g}, \boldsymbol{y}^T\boldsymbol{g}) +$$
$$\boldsymbol{r}^T\boldsymbol{g}s_p^r(\boldsymbol{x})\nabla_{\boldsymbol{y}^T\boldsymbol{g}}k_{rg}(\boldsymbol{x}^T\boldsymbol{g}, \boldsymbol{y}^T\boldsymbol{g}) + (\boldsymbol{r}^T\boldsymbol{g})^2\nabla^2_{\boldsymbol{x}^T\boldsymbol{g}, \boldsymbol{y}^T\boldsymbol{g}}k_{rg}(\boldsymbol{x}^T\boldsymbol{g}, \boldsymbol{y}^T\boldsymbol{g}). \quad (8)$$

The following theorem describes the optimal test function inside SSD (Eq.(5)) and maxSSD (Eq.(6)).

**Theorem 2.** *(Closed form solution) If $\mathbb{E}_q[h_{p,r,g}(\boldsymbol{x}, \boldsymbol{x})] < \infty$, then*

$$D_{rg}^2(q,p) = ||\sup_{f_{rg} \in \mathcal{H}_{rg}, ||f_{rg}|| \leq 1} \mathbb{E}_q[s_p^r(\boldsymbol{x})f_{rg}(\boldsymbol{x}^T\boldsymbol{g}) + \boldsymbol{r}^T\boldsymbol{g}\nabla_{\boldsymbol{x}^T\boldsymbol{g}}f_{rg}(\boldsymbol{x}^T\boldsymbol{g})]||^2$$
$$= ||\mathbb{E}_q[\xi_{p,r,g}(\boldsymbol{x})]||^2_{\mathcal{H}_{rg}} = \mathbb{E}_{q(\boldsymbol{x})q(\boldsymbol{x}')}[h_{p,r,g}(\boldsymbol{x}, \boldsymbol{x}')]. \quad (9)$$

Next, we propose the kernelized version of SSD with orthogonal basis $O_r$, called SKSD.

**Theorem 3.** *(SKSD as a discrepancy) For two probability distributions $p$ and $q$, given assumptions 1,2 and 5 in appendix A and $\mathbb{E}_q[h_{p,r,g}(\boldsymbol{x}, \boldsymbol{x})] < \infty$ for all $\boldsymbol{r}$ and $\boldsymbol{g}$, we define SKSD as*

$$SK_o(q,p) = \sum_{\boldsymbol{r} \in O_r} \int_{\mathbb{S}^{D-1}} p_g(\boldsymbol{g})D_{rg}^2(q,p)d\boldsymbol{g}, \quad (10)$$

*which is equal to 0 if and only if $p = q$ a.e.*

Following the same idea of maxSSD (Eq.6), it suffices to use optimal slice direction $\boldsymbol{g}_r$ for each $\boldsymbol{r} \in O_r$, resulting in a slicing matrix $\boldsymbol{G} \in \mathbb{S}^{D \times (D-1)}$. We name this discrepancy as maxSKSD, or *maxSKSD-g* when we need to distinguish it from another variant described later.

**Corollary 3.1.** *(maxSKSD) Assume the conditions in Theorem 3 are satisfied. Then*

$$SK_{max}(q,p) = \sum_{\boldsymbol{r} \in O_r} \sup_{\boldsymbol{g}_r} D_{rg_r}^2(q,p) \quad (11)$$

*is equal to 0 if and only if $p = q$ a.e.*

Figure 1 (Left) clarifies the connections between the mentioned discrepancies. We emphasise that using a single projection $\boldsymbol{g}$ in maxSKSD may be insufficient when no single projected feature $\boldsymbol{x}^T\boldsymbol{g}$ is informative enough to describe the difference between $p$ and $q$. Instead, in maxSKSD, for each score projection $\boldsymbol{r} \in O_r$, we have a corresponding $\boldsymbol{g}_r$. One can also use the optimal $\boldsymbol{r}$ to replace the summation over $O_r$, which provides additional benefits in certain GOF tests. We call this discrepancy *maxSKSD-rg*, and its validity can be proved accordingly. Interestingly, in appendix G, we show under certain scenarios *maxSKSD-g* can have inferior performance due to the noisy information provided by the redundant dimensions. Further, we show that such limitation can be efficiently addressed by using *maxSKSD-rg*.

**Kernel choice and optimal $G$** RBF kernel with median heuristics is a common choice. However, better kernels, e.g. deep kernels which evaluate a given kernel on the transformed input $\phi(\boldsymbol{x})$, might be preferred. It is non-trivial to directly use such kernel on SKSD or maxSKSD. We propose an adapted form of Eq.(10) to incorporate such kernel and maintain its validity. We include the details in appendix D and leave the experiments for future work.

The quality of sliced direction $G$ is crucial for the performance of both *maxSKSD-g* or *maxSKSD-rg*. Indeed, it represents the projection directions that two distributions differ the most. The closed-form solutions of $G$ is not analytic in general, in practice, finding the optimal $G$ involves solving other difficult optimizations as well (projection $r$ and test function $f_{rg}$). For the scope of this work, we obtained $G$ by optimizing *maxSKSD-g* or *maxSKSD-rg* using standard gradient optimization, e.g. Adam, with random initialization. Still in some special cases (e.g. $p, q$ are full-factorized), analytic solutions of optimal $G$ exists, which is further discussed in appendix E.

### 3.3 APPLICATION OF MAXSKSD

**Goodness-of-fit Test** Assume the optimal test directions $\boldsymbol{g}_r \in \boldsymbol{G}$ are available, maxSKSD (Eq.(11)) can then be estimated using U-statistics (Hoeffding, 1992; Serfling, 2009). Given i.i.d. samples $\{\boldsymbol{x}_i\}_{i=1}^N \sim q$, we have an unbiased minimum variance estimator:

$$\widehat{SK}_{max}(q,p) = \frac{1}{N(N-1)} \sum_{\boldsymbol{r} \in O_r} \sum_{1 \leq i \neq j \leq N} h_{p,r,g_r}(\boldsymbol{x}_i, \boldsymbol{x}_j). \tag{12}$$

The asymptotic behavior of the estimator is analyzed in appendix F.1. We use bootstrap (Liu et al., 2016; Huskova & Janssen, 1993; Arcones & Gine, 1992) to determine the threshold for rejecting the null hypothesis as indicated in algorithm 1. The bootstrap samples can be calculated by

$$\widehat{SK}_m^* = \sum_{1 \leq i \neq j \leq N} (w_i^m - \frac{1}{N})(w_j^m - \frac{1}{N}) \sum_{\boldsymbol{r} \in O_r} h_{p,r,g_r}(\boldsymbol{x}_i, \boldsymbol{x}_j) \tag{13}$$

where $(w_1^m, \ldots, w_N^m)_{m=1}^M$ are random weights drawn from multinomial distributions $\text{Multi}(N, \frac{1}{N}, \ldots, \frac{1}{N})$.

---

**Algorithm 1:** GOF Test with maxSKSD U-statistics

**Input** : Samples $\{\boldsymbol{x}_i\}_{i=1}^N \sim q(\boldsymbol{x})$, score function $\boldsymbol{s}_p(\boldsymbol{x})$, Orthogonal basis $O_r$, optimal test direction $\boldsymbol{g}_r$ for each $\boldsymbol{r} \in O_r$, kernel function $k_{rg}$, significant level $\alpha$, and bootstrap sample size $M$.

**Hypothesis :** $H_0$: $p = q$ v.s. $H_1$: $q \neq p$

Compute $\widehat{SK}_{max}(q,p)$ using U-statistic Eq.(12);

Generate $M$ bootstrap samples $\{\widehat{SK}_m^*\}_{m=1}^M$ using Eq.(13);

Reject null hypothesis $H_0$ if the proportion $\widehat{SK}_m^* > \widehat{SK}_{max}(q,p)$ is less than $\alpha$.

---

**Model Learning** The proposed maxSKSD can be applied to model learning in two ways. First, it can be directly used as a training objective, in such case $q$ is the data distribution and $p$ is the model to be learned, and the learning algorithm performs $\min_p SK_{max}(q,p)$. The second model learning scheme is to leverage the particle inference for latent variables and train the model parameters using an EM-like (Dempster et al., 1977) algorithm. Similar to the relation between SVGD and KSD, we can derive a corresponding particle inference algorithm based on maxSKSD, called *sliced-SVGD* (S-SVGD). In short, we define a specific form of the perturbation as $\phi(\boldsymbol{x}) = [\phi_{g_i}(\boldsymbol{x}^T \boldsymbol{g}_i), \ldots, \phi_{g_D}(\boldsymbol{x}^T \boldsymbol{g}_D)]^T$ and modify the proofs of Lemma 1 accordingly. The resulting S-SVGD algorithm uses kernels defined on one dimensional projected samples, which sidesteps the vanishing repulsive force problem of SVGD in high dimensions (Zhuo et al., 2017; Wang et al., 2018). We illustrate this in Figure 2 by estimating the variance

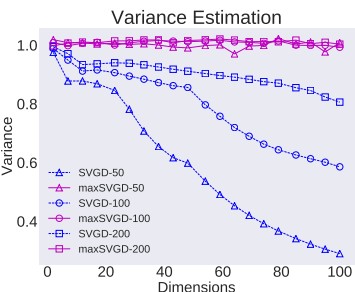

Figure 2: Estimating the average variance of $p(\boldsymbol{x}) = \mathcal{N}(\boldsymbol{0}, \boldsymbol{I})$ across dimensions using SVGD particles. SVGD-50 means the variance are estimated using 50 samples.

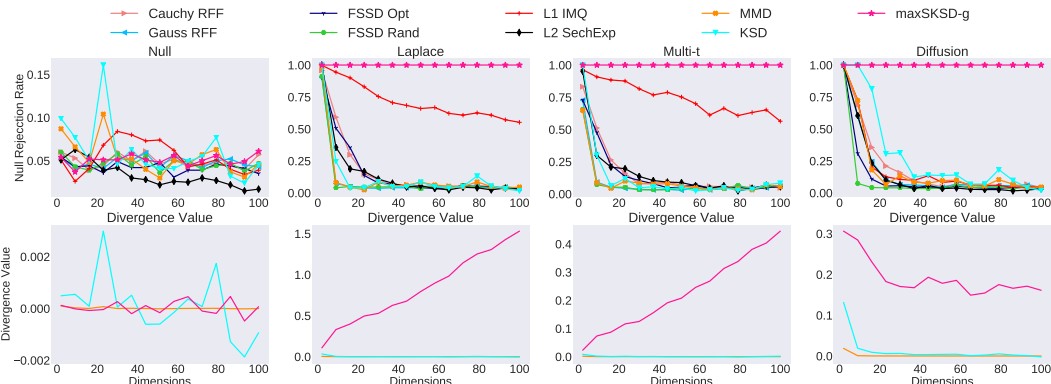

Figure 3: Each column reports GOF test results for a different alternative hypothesis, with the upper panel showing the rejection rate of the Null hypothesis and the lower panel showing the discrepancy value averaged over all trials. Both quantities are plotted w.r.t. the number of dimensions.

of a standard Gaussian with the particles obtained by SVGD or S-SVGD (see appendix J.1). We see that as the dimension increases, SVGD severely under-estimates the variance of $p$, while the S-SVGD remains robust. Furthermore, its validity is justified since in such case the KL gradient equals to maxSKSD which is a valid discrepancy. Readers are referred to appendix F.2 for the derivations. We also give an analysis of their memory and computational cost for both GOF and model learning in appendix H.

## 4 EXPERIMENTS

### 4.1 GOODNESS OF FIT TEST

We evaluate maxSKSD (Eq.(11)) for GOF tests in high dimensional problems. First, we demonstrate its robustness to the increasing dimensionality using the Gaussian GOF benchmarks (Jitkrittum et al., 2017; Huggins & Mackey, 2018; Chwialkowski et al., 2016). Next, we show the advantage of our method for GOF tests on 50-dim *Restricted Boltzmann Machine* (RBM) (Liu et al., 2016; Huggins & Mackey, 2018; Jitkrittum et al., 2017). We included in comparison extensive baseline test statitics for GOF test: Gaussian or Cauchy random Fourier features (RFF) (Rahimi & Recht, 2008), KSD with RBF kernel (Liu et al., 2016; Chwialkowski et al., 2016), finite set Stein discrepancy (FSSD) with random or optimized test locations (Jitkrittum et al., 2017), random feature Stein discrepancy (RFSD) with L2 SechExp and L1 IMQ kernels (Huggins & Mackey, 2018), and maximum mean discrepancy (MMD) (Gretton et al., 2012) with RBF. Notice that we use gradient descent to obtain the test directions $\boldsymbol{g}_r$ (and potentially the slicing directions $\boldsymbol{r}$) for Eq.(11).

### 4.1.1 GOF TESTS WITH HIGH DIMENSIONAL GAUSSIAN BENCHMARKS

We conduct 4 different benchmark tests with $p = \mathcal{N}(0, \boldsymbol{I})$: (1) **Null test**: $q = p$; (2) **Laplace**: $q(\boldsymbol{x}) = \prod_{d=1}^{D} \text{Lap}(x_d | 0, 1/\sqrt{2})$ with mean/variance matched to $p$; (3) **Multivariate-t**: $q$ is fully factorized multivariate-t with 5 degrees of freedom, 0 mean and scale 1. In order to match the variance of $p$ and $q$, we change the variance of $p$ to $\frac{5}{5-2}$; (4) **Diffusion**: $q(\boldsymbol{x}) = \mathcal{N}(\boldsymbol{0}, \boldsymbol{\Sigma}_1)$ where the variance of $1^{\text{st}}$-dim is 0.3 and the rest is the same as in $\boldsymbol{I}$. For the testing setup, we set the significance level $\alpha = 0.05$. For FFSD and RFSD, we use the open-sourced code from the original publications. We only consider maxSKSD-g here as it already performs nearly optimally. We refer to appendix I.1 for details.

Figure 3 shows the GOF test performances and the corresponding discrepancy values. In summary, the proposed maxSKSD outperforms the baselines in all tests, where the result is robust to the increasing dimensions and the discrepancy values match the expected behaviours.

**Null** The left-most column in Figure 3 shows that all methods behave as expected as the rejection rate is closed to the significance level, except for RFSD with L2 SechExp kernel. All the discrepancy values oscillate around 0, with the KSD being less stable.

**Laplace and Multivariate-t** The two middle columns of Figure 3 show that maxSKSD-g achieves a nearly perfect rejection rate consistently as the dimension increases, while the test power for all

Table 1: Test NLL for different dimensional ICA with different objective functions. The above results are averaged over 5 independent runs of each methods.

| Method | Dimension | | | | | | |
|---|---|---|---|---|---|---|---|
| | $D = 10$ | $D = 20$ | $D = 40$ | $D = 60$ | $D = 80$ | $D = 100$ | $D = 200$ |
| KSD | **-10.23** | -15.98 | -34.50 | -56.87 | -86.09 | -116.51 | -329.49 |
| LSD | -10.42 | -14.54 | **-17.16** | **-15.05** | -12.39 | -5.49 | 46.63 |
| maxSKSD | -10.45 | **-14.50** | -17.28 | -15.70 | **-11.91** | **-4.21** | **47.72** |

baselines decreases significantly. For the discrepancy values, similar to the KL divergence between $q$ and $p$, maxSKSD-g linearly increases with dimensions due to the independence assumptions..

**Diffusion**    This is a more challenging setting since $p$ and $q$ only differ in one of their marginal distributions, which can be easily buried in high dimensions. As shown in the rightmost column of Figure 3, all methods failed in high dimensions except maxSKSD-g, which still consistently achieves optimal performance. For the discrepancy values, we expect a positive constant due to the one marginal difference between $p$ and $q$. Only maxSKSD-g behaves as expected as the problem dimension increases. The decreasing value at the beginning is probably due to the difficulty in finding the optimal direction $\boldsymbol{g}$ in high dimensions when the training set is small.

### 4.1.2   RBM GOF TEST

We demonstrate the power maxSKSD for GOF tests on RBMs, but we now also include results for *maxSKSD-rg*. We follow the test setups in Liu et al. (2016); Jitkrittum et al. (2017); Huggins & Mackey (2018) where different amounts of noise are injected into the weights to form the alternative hypothesis $q$. The samples are drawn using block Gibbs samplers. Refer to appendix I.2 for details. Figure 4 shows that maxSKSD based methods dominate the baselines, especially with maxSKSD-rg significantly outperforming the others. At perturbation level $0.01$, maxSKSD-rg achieves $0.96$ rejection rate, while others are all below $0.5$. This result shows the advantages of optimizing the slicing directions $\boldsymbol{r}$.

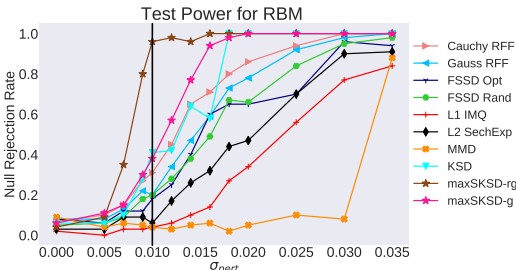

Figure 4: RBM GOF Test with different levels of perturbation noise. The black vertical line indicates the perturbation level at $0.01$.

### 4.2   MODEL LEARNING

We evaluate the efficiency of maxSKSD-based algorithms in training machine learning models. First, we use *independent component analysis* (ICA) which is often used as a benchmark for evaluating training methods for energy-based model (Gutmann & Hyvärinen, 2010; Hyvärinen, 2005; Ceylan & Gutmann, 2018). Our approach trains the ICA model by directly minimizing maxSKSD. Next, we evaluate the proposed S-SVGD particle inference algorithm, when combined with *amortization* (Feng et al., 2017; Pu et al., 2017), in the training of a *variational autoencoder* (VAE) (Kingma & Welling, 2013; Rezende et al., 2014) on binarized MNIST. Appendix J.5 also shows superior results for S-SVGD when training a Bayesian neural network (BNN) on UCI datasets (Dua & Graff, 2017).

### 4.2.1   ICA

ICA consists of a simple generative process $\boldsymbol{z} \sim \mathrm{Lap}(0, 1)$ and $\boldsymbol{x} = \boldsymbol{W}\boldsymbol{z}$, where the model parameters are a non-singular matrix $\boldsymbol{W} \in \mathbb{R}^{D \times D}$. The log density for $\boldsymbol{x}$ is $\log p(\boldsymbol{x}) = \log p_z(\boldsymbol{W}^{-1}\boldsymbol{x}) + C$, where the normalization constant $C$ can be ignored when training with Stein discrepancies. We train the models on data sampled from a randomly initialized ICA model and evaluate the corresponding test log likelihoods. We compare maxSKSD with KSD and the state-of-the-art LSD (Grathwohl et al., 2020). For more details on the setup, we refer the reader to appendix J.2.

Table 1 shows that both maxSKSD and LSD are robust to increasing dimensions, with maxSKSD being better when $D$ is very large. Also at $D = 200$, maxSKSD converges significantly faster than LSD (see Figure 10 in appendix J.3). This faster convergence is due to the closed-form solution for the optimal test functions, whereas LSD requires adversarial training. While KSD is also kernel-based, it suffers from the curse-of-dimensionality and fails to train the model properly for $D > 20$. Instead the proposed maxSKSD can successfully avoid the problems of KSD with high dimensional data.

Table 2: Average log likelihood on first $5,000$ test images for different $D$ of latent dimensions.

| Method | Latent Dim | | | |
|---|---|---|---|---|
| | D=16 | D=32 | D=48 | D=64 |
| Vanilla VAE | -91.50 | -90.39 | -90.58 | -91.50 |
| SVGD VAE | **-88.58** | -90.43 | -93.47 | -94.88 |
| S-SVGD VAE | -89.17 | **-87.55** | **-87.74** | **-87.78** |

Table 3: Label entropy and accuracy for imputed images.

| Method | Entropy | Accuracy |
|---|---|---|
| Vanilla VAE | 0.297 | 0.718 |
| SVGD VAE | 0.538 | 0.691 |
| S-SVGD VAE | **0.542** | **0.728** |

#### 4.2.2 AMORTIZED SVGD

Finally, we consider training VAEs with implicit encoders on dynamically binarized MNIST. The decoder is trained as in vanilla VAEs, but the encoder is trained by amortization (Feng et al., 2017; Pu et al., 2017), which minimizes the mean square error between the initial samples from the encoder, and the modified samples driven by the SVGD/S-SVGD dynamics (Algorithm 3 in appendix J.4).

We report performance in terms of test log-likelihood (LL). Furthermore we consider an imputation task, by removing the pixels in the lower half of the image and imputing the missing values using (approximate) posterior sampling from the VAE models. The performance is measured in terms of imputation diversity and correctness, using label entropy and accuracy. For fair comparisons, we do not tune the coefficient of the repulsive force. We refer to appendix J.4 for details.

Table 2 reports the average test LL. We observe that S-SVGD is much more robust to the increasing latent dimensions compared to SVGD. To be specific, with $D = 16$, SVGD performs the best where S-SVGD performs slightly worse than SVGD. However, when the dimension starts to increase, LL of SVGD drops significantly. For $D = 64$, a common choice for latent space, it performs even significantly worse than vanilla VAE. On the other hand, S-SVGD is much more robust. Notice that the purpose of this experiment is to show compare their robustness instead of achieving the state-of-the-art performance. Still the performance can be easily boosted, e.g. running longer S-SVGD steps before encoder update, we leave it for the future work.

For the imputation task, we compute the label entropy and accuracy for the imputed images (Table 3). We observe S-SVGD has higher label entropy compared to vanilla VAE and better accuracy compared to SVGD. This means both S-SVGD and SVGD capture the muli-modality nature of the posterior compared to uni-modal Gaussian distribution. However, high label entropy itself may not be a good indicator for the quality of the learned posterior. One can think of a counter-example that the imputed images are diverse but does not look like any digits. This may also gives a high label entropy but the quality of the posterior is poor. Thus, we use the accuracy to indicate the "correctness" of the imputed images, with higher label accuracy meaning the imputed images are closed to the original image. Together, a good model should give a higher label entropy along with the high label accuracy. We observe S-SVGD has more diverse imputed images with high imputation accuracy.

### 4.3 SUMMARY OF THE EXPERIMENTS IN APPENDIX

We present further empirical results on GOF tests and model learning in the appendix to demonstrate the advantages of the proposed maxSKSD. As a summary glance of the results:

- In appendix G, we analyse the potential limitations of maxSKSD-g and show that they can be mitigated by maxSKSD-rg, i.e. optimising the slicing direction $r$;
- In appendix I.3, we successfully apply maxSKSD to selecting the step size for *stochastic gradient Hamiltonian Monte Carlo* (SGHMC) (Chen et al., 2014);
- In appendix J.5, we show that the proposed S-SVGD approach out-performs the original SVGD on Bayesian neural network regression tasks.

## 5 RELATED WORK

**Stein Discrepancy** SD (Gorham & Mackey, 2015) and KSD (Liu et al., 2016; Chwialkowski et al., 2016) are originally proposed for GOF tests. Since then research progress has been made to improve these two discrepancies. For SD, LSD (Grathwohl et al., 2020; Hu et al., 2018) is proposed to increase the capacity of test functions using neural networks with $L_2$ regularization. On the other hand, FSSD (Jitkrittum et al., 2017) and RFSD (Huggins & Mackey, 2018) aim to reduce

the computation cost of KSD from $O(n^2)$ to $O(n)$ where $n$ is the number of samples. Still the curse-of-dimensionality issue remains to be addressed in KSD, and the only attempt so far (to the best of our knowledge) is the *kernelized complete conditional Stein discrepancy* (KCC-SD (Singhal et al., 2019)), which share our idea of avoiding kernel evaluations on high dimensional inputs but through comparing conditional distributions. KCC-SD requires the sampling from $q(x_d|\boldsymbol{x}_{-d})$, which often needs significant approximations in practice due to its intractability. This makes KCC-SD less suited for GOF test due to estimation quality in high dimensions. On the other hand, our approach does not require this approximation, and the corresponding estimator is well-behaved asymptotically.

**Wasserstein Distance and Score matching**    Sliced Wasserstein distance (SWD) (Kolouri et al., 2016) and sliced score matching (SSM) (Song et al., 2019) also uses the "slicing" idea. However, their motivation is to address the computational issues rather than statistical difficulties in high dimensions. SWD leveraged the closed-form solution of 1D Wasserstein distance by projecting distributions onto 1D slices. SSM uses Hutchson's trick (Hutchinson, 1990) to approximate the trace of Hessian.

**Particle Inference**    Zhuo et al. (2017); Wang et al. (2018) proposed *message passing SVGD* to tackle the well-known mode collapse problem of SVGD using local kernels in the graphical model. However, our work differs significantly in both theory and applications. Theoretically, the discrepancy behind their work is only valid if $p$ and $q$ have the same Markov blanket structure (refer to Section 3 in Wang et al. (2018) for detailed discussion). Thus, unlike our method, no GOF test and practical inference algorithm can be derived for generic cases. Empirically, the Markov blanket structure information is often unavailable, whereas our method only requires projections that can be easily obtained using optimizations. *Projected SVGD* (pSVGD) is a very recent attempt (Chen & Ghattas, 2020) which updates the particles in an adaptively constructed low dimensional space, resulting in a biased inference algorithm. The major difference compared to S-SVGD is that our work still updates the particles in the original space with kernel being evaluated in 1D projections. Furthermore, S-SVGD can theoretically recover the correct target distribution. There is no real-world experiments provided in (Chen & Ghattas, 2020), and a stable implementation of pSVGD is non-trivial, so we did not consider pSVGD when selecting the baselines.

## 6    CONCLUSION

We proposed sliced Stein discrepancy (SSD), as well as its scalable and kernelized version maxSKSD, to address the curse-of-dimensionality issues in Stein discrepancy. The key idea is to project the score function on one-dimensional slices and define (kernel-based) test functions on one-dimensional projections. We also theoretically prove their validity as a discrepancy measure. We conduct extensive experiments including GOF tests and model learning to show maxSKSD's improved performance and robustness in high dimensions. There are three exciting avenues of future research. First, although validated by our theoretical study in appendix D, practical approaches to incorporate deep kernels into SSD remains an open question. Second, the performance of maxSKSD crucially depends on the optimal projection direction, so better optimization methods to efficiently construct this direction is needed. Lastly, we believe "slicing" is a promising direction for kernel design to increase the robustness to high dimensional problems in general. For example, MMD can be easily extended to high dimensional two-sample tests using this kernel design trick.

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
