# OpenReview forum: "Sliced Kernelized Stein Discrepancy"
_ICLR.cc/2021/Conference — ICLR 2021 Poster_

### Official Review · AnonReviewer3 · 2020-10-27
**Interesting work, the proposed method seems to address several known limitations of the KSD and SVGD**

**Rating:** 8
**Confidence:** 4

**Review:**



The paper proposes a new discrepancy between probability distributions called the sliced Kernelized Stein Discrepancy.  It is based on the idea of computing the standard KSD on random 1 dimensional projections and then average those. Some other proposed variants are based on optimizing over the projections.

The motivation behind this new discrepancy is to overcome the 'curse of dimensionality' that standard KSD suffers from. Thus the main selling point is that the sliced version has a much better behavior as the dimension increases.

The authors then propose to use the new divergence for two applications: goodness of fit testing and for models learning.

The robustness to increasing dimensions is illustrated through multiple experiments on both tasks and yields convincing results.

Strength:
	- The proposed framework is neat and the experiments are thorough and convincing with many additional discussions and results in the appendix.
	- The proposed method is relatively easy to implement and seems to address several known limitations of KSG and in SVGD: the scaling with dimensions.



Weaknesses:
 The paper is a bit dense with many references to the appendix. However the main idea is clearly explained  and the advantage is clear both in terms of theory and throughout the experiments.


Questions:
	- In 4.1.1, the distributions p and q although high dimensional, they often have independent components. This might be very advantageous for the sliced version of the algorithm, especially when using a set of orthogonal projections for the projections $r$. What happens to the Null rejection rate when more dependence between the dimensions is introduced ? What is the exact parameter choice for the multivariate-t distribution, I couldn't find this in the appendix?

 Minor remarks.
 - A discussion on the limitations of existing methods KSD and SVGD could be useful as a transition from section 2 to 3 to motivate the slicing.
- Figure 1 is a bit dense especially with all the equations
- The subscript notation $f_{rg}$ is sometimes confusing as f depends on $r$ and $g$ only implicitly after optimizing the objective in 5. It might be worth either mentioning where this dependence comes from or even remove it.
- In the paragraph right after corollary 3.1. The authors mention a limitation of a particular version of Max Sliced KSD over the other but then refer to appendix F without really saying what this limitation is. It might be worth saying a little bit more about those limitations.

---

> ### Author Response · Authors · 2020-11-20
> **Author's Response**
>
> Thank you for your valuable suggestions to improve the paper and positive opinions for our work. In the following, we will try to address each concern in the review.
>
> 1. The null rejection rate for distributions with high correlations between dimensions.
> We conduct the goodness-of-fit test using RBM in section 4.1.2, which is a complex multi-modal distribution with correlated dimensions. When the weight matrix perturbation magnitude is 0, $p$=$q$. From figure 4, we can observe the null-rejection rate is closed to the significant level $5$% ($6.32$% for maxSKSD-g and $4.78$% for maxSKSD-rg to be exact). When $p\neq q$ (i.e. with non-zero perturbation), our method is much more sensitive and consistently has higher power than baselines.
> 2. Parameters for multivariate-t distribution.
> We include the detailed parameters in the revised version. To be specific, we use fully factorized multivariate-t distribution (i.e. for each dimension, it is a student-t distribution). We use $0$ mean and $1$ scale with $5$ degree of freedom. In order to match the variance of $p$ (Gaussian) and $q$ (multivariate-t), we set the variance of $p$ to be $\frac{5}{5-2}$.
> 3. The confusing notation of $f_{rg}$.
> We add some comments about this in section 3.1 of the revised version. One should note that $\pmb{r}$ and $\pmb{g}$ should not be treated just as the parameters for test function $f$. Instead, they are more like the index indicating that for each pair of directions $\pmb{g}$, $\pmb{r}$, one need a new test function $f_{rg}$ (obtained by solving the supremum in Eq.5 for each pair of $\pmb{r}$,$\pmb{g}$)
> 4. Limitations of maxSKSD-g
> We modify the bottom of page 4 by directly stating its limitations. In a nutshell, maxSKSD-g can give an inferior performance under certain scenarios due to noisy redundant information provided by 'unimportant' basis. This increased variance can hurt the test power of maxSKSD-g.

---

### Official Review · AnonReviewer2 · 2020-10-27
**Details of the optimal test direction are needed**

**Rating:** 6
**Confidence:** 4

**Review:**

This paper tries to solve the curse-of-dimensionality problem of KSD and corresponding mode-collapse problem of SVGD by projecting both the input and output of test function onto 1D slices. By doing so, the paper proposes the new discrepancies called SSD and maxSKSD, and a new variant of SVGD called S-SVGD. Experiments on goodness-of-fit test (synthetic high-dim Gaussian & RBM) and model learning (ICA on synthetic data & amortized SVGD on MNIST) are reported in the main body of the paper.

This paper is well motivated and the writing is good. Curse-of-dimensionality problem is common in kernel based methods like KSD and corresponding SVGD. To the best of my knowledge, the idea of 'slicing' is novel to solve the high dimensional problem of KSD and SVGD. However, the derivation and analysis (e.g., computational complexity, whether closed form or not, how to get it in each experiment, etc...) of the optimal test direction $g_r$ are missed in the main body of the paper, which is critical to evaluate the quality of the proposed maxSKSD and S-SVGD according to my understanding.

Pros:
1. The 'slicing' idea is novel to KSD and SVGD;
2. The paper is easy to follow in general, though some minor points need further improvement;
3. Experiments are conducted on various tasks and datasets, which provides a thorough comparison.

Cons:
1. Lack of analysis of the optimal test direction $g_r$. According to Eq. (6), it seems that $g_r \in R^D$ has no closed-form solution and corresponding $G \in R^{D\times D}$ in general.This raises several questions:

     1.1 In the high dimensional case (D >> 1),  $D\times D$ seems unacceptable. Is it possible to provide the empirical storage consumption of both S-SVGD and SVGD in experiments, especially for neural networks?

     1.2 Getting $g_r$ requires solving a maximization problem (see Algorithm 2 in the appendix), which introduces an additional inner loop for each iteration of S-SVGD. So what is the time complexity? Is it possible to provide the comparison of wall clock time of S-SVGD and SVGD in experiments?

     1.3 Does $g_r$ have closed-form solution in some special case, e.g., Gaussian?

2. The notation $f$ is used for denoting both $R^D\to R^D$ and $R^D \to R$ mapping, which is misleading. $f(\cdot;r,g):R^D \to R$ and $f_{rg}:R\to R$ are even more confusing. Besides, it is a little hard to understand how Eq. (5) can be derived based on Eq. (2).

---

> ### Author Response · Authors · 2020-11-20
> **Author's Response**
>
> Thank you for your valuable suggestions to improve the paper. In the following, we will try to address each concern in the review.
> 1. How to select $\pmb{G}$ and its closed form?
> We add a detailed discussion on how to select the optimal slicing direction $\pmb{G}$ and its optimal form in the beginning of page 5 and Appendix E in the revised version of the paper. For short answer, refer to the official comments "Revision of the paper".
> 2. Memory and computational cost of the proposed method.
> We include a detailed discussion along with the empirical memory consumption and wall-clock time for the BNN experiment in Appendix H. We admit that the proposed method indeed consumes more memory and requires more time to compute compared to KSD or SVGD. The computational and memory cost of our methods (maxSKSD-g and S-SVGD) is $O(N^2D^2)$ v.s. $O(N^2D)$ for KSD and SVGD, where $N$ is the number of samples and $D$ is the dimension. However, we discuss that such cost can be reduced in GOF test by using maxSKSD-rg. The cost can be reduced to $O(N^2Dm)$ where $m$ is the number of important basis selected by optimizing $\pmb{r}$, and normally $m\ll D$. In practice, for the BNN experiment, the memory cost is 1003MB and 1203MB for SVGD and S-SVGD respectively. SVGD uses 0.032s per epoch whereas S-SVGD uses 0.073s.  However, in the amortized SVGD experiment, the time consumptions are 0.112s and 0.122s per iteration for SVGD and S-SVGD respectively.
> We should stress that there is no free lunch and the significant advantages of our method come with the price of higher computation and memory cost. **But this does not hinder its applicability much. One should note that the curse-of-dimensionality happens very early for KSD/SVGD (starts at around dimension 20-30).**  We can observe this through the GOF test and amortized SVGD experiment. Thus, under this scenario, our method can be easily applied without heavy memory/computational burden for moderately high dimensions where other variants of KSD/SVGD completely fail. **In addition, we argue that S-SVGD for BNN is only one of the applications of the proposed framework. In other application such as GOF test, maxSKSD-rg does not have high memory/computation cost compared to KSD.**
> 3. We slightly modifies the notations for function $f$ in section 3.1 and add some comments to help the understanding. In fact, the function $\pmb{f}:\mathbb{R}^D\rightarrow \mathbb{R}^D$ is only used for Stein discrepancy (Eq. 2). For our proposed method, the $f(\cdot;\pmb{r},\pmb{g}):\mathbb{R}^D\rightarrow \mathbb{R}$ is a new test function that is different from the one used in Stein discrepancy. We define $f(\cdot;\pmb{r},\pmb{g})=f_{rg}\circ h_g(\pmb{x})$ where $h_g(\pmb{x})=\langle\pmb{x},\pmb{g}\rangle$. Thus, the $f_{rg}$ is a $\mathbb{R}\rightarrow\mathbb{R}$ function.
> In addition, we did not use the Stein discrepancy (Eq.2) to obtain Eq.5. However, one can indeed recover Eq.5 using Eq.2 by selecting a special form of test function for Stein discrepancy. The details are in the proof of Proposition 1 (Appendix B.2).
>
> Hope this addresses the raised concerns, we humbly hope that you could reconsider the score given this response and the additional materials in the revised version of our paper.

---

> > ### Comment · AnonReviewer2 · 2020-11-23
> > **One more question**
> >
> > Thanks for your response. It addresses most of my concerns and I raise my score accordingly.
> > For the BNN experiment, since G has no closed-form solution for NN and requires finding the optimal one through gradient descent per iteration (Algorithm 2 in the appendix), I'm curious about the total running time of S-SVGD compared to SVGD (i.e., the wall clock time of L iterations instead of per iteration). Besides, usually how many iterations are conducted for getting the optimal G in Algorithm 2?

---

> > > ### Author Response · Authors · 2020-11-24
> > > **Totoal running time of S-SVGD and SVGD**
> > >
> > > Thank you so much for the updated score, and I am glad that the initial response addressed most of the concerns.
> > >
> > > The time reported in the initial response is the time per epoch (including the inner loop optimization). For the Boston housing data set, we run 2000 epochs to make sure both algorithms converged. The total running time are 154.72s for S-SVGD and 66.25s for SVGD.
> > >
> > > As for the inner loop optimization, we only update $\pmb{G}$ once per iteration, which is often adopted in the adversarial training procedure. We tested longer inner loop optimization (update 5 times per iteration), it does not seem to produce significantly better results. So take the computational complexity into account, we decided to update once per iteration.

---

### Official Review · AnonReviewer4 · 2020-10-28
**Proposed sliced kernelized stein discrepancy, which is useful for high dimensional models.**

**Rating:** 6
**Confidence:** 3

**Review:**

##  Summary of the paper
The authors proposed the `sliced version of the kernelized stein discrepancy and solved the collapsed problem for high dimensional GOF and particle based model learning.

## Strong and weak points of the paper
### Strong points
- Provided a novel slicing technique for KSD and provided its statistical estimator based on U-static.
- Experimental results supports that the proposed sliced KSD is very promising approach for high dimensional models.

### Weak points (Questions)
- I think the presentation should be modified so that the reader can easily understand the proposed methods. At least, the main algorithms should be presented in the main paper. For example, the algorithm for GOF test is not shown in the main paper but included in Appendix E, although it is the main algorithm of this work.
- I could not understand when I should use MAXSKSD-G, not MAXSKSD-RG although detailed discussion is shown in Appendix F. To me, based on the discussion of Appendix F, MAXSKSD-RG seems always be better than MAXSKSD-G. So don't we need MAXSKSD-G?
- (I might overlooked, but) the strategy on how to choose optimal G for GOF test is not shown in the main paper although it is expained in Appendix G. The explanation of how to tune G should be included in the main paper.

## Rating
- Clarity: For me, the main paper is not enough to understand the paper and the proposed method clearly. I needed to read the Appendix carefully.
- Correctness: I did not check the proofs of each lemma in Appendix B.
- Novelty: The idea seems very interesting and important in the community.

## Comments and Questions
- Comments) I think, instead of Figure 1,  Appendix B.1 should be included in the main paper, which is very helpful to understand the overall strategy of the proposed discrepancy.

- Minor comments )
I think Eq numbers in Figure 1 Left seems wrong, especially for SSD and max SSD.
On page 17, appendix  B.2, the sentence above Eq.29 says, ``"~~~ as in Theorem 3~~~", I think this is a typo and should be replaced with "Lemma 3".

---

> ### Author Response · Authors · 2020-11-20
> **Author's response**
>
> Thank you for your valuable suggestions to improve the paper. In the following, we will try to address each concern in the review.
> 1. Presentation should be modified.
> We agree the paper is a bit dense to read without referring to the appendix. To improve readability, we add the GOF algorithm and equations to compute bootstrap samples in the main text. In addition, we also add some comments about the intuition behind the proposed sliced Stein discrepancy in section 3.1.
> 2. When maxSKSD-g is used over maxSKSD-rg?
> We briefly answer this question in the official comments "Revision of the paper". We include a detailed discussion in Appendix G.4 with an extra experiment to support our claim.
> 3. See the official comments "Revision of the paper"
> 4. We slightly modified section 3.1 by moving some contents in Appendix B.1 to the main text, especially the intuition related to Radon transform.
> 5. Thank you for noticing the typos, we changed it in the revised version.
>
> Hope this addresses the raised concerns, we humbly hope that you could reconsider the score given the additional material in the revised version of our paper.

---

### Author Response · Authors · 2020-11-20
**Revision of the paper**

We thank all the reviewers for their valuable time and detailed comments towards improving the paper. According to the suggestions, we made several modifications in this revised version to improve the readability and address the concerns raised by the reviewers. The modifications are indicated using red colour in the revision. Here is a brief summary of the major changes:
* Slightly changes section 3.1 to improve the readability of the intuition behind sliced Stein divergence.
* Add a discussion on how to obtain slicing direction $\pmb{G}$ in our paper. In addition, we theoretically analyzed its closed-form solution under the condition that $p$ and $q$ are fully factorized. The details are included in the beginning of page 5 and Appendix E.
* To address the concerns raised by Reviewer 2, we add a discussion on the computational and memory cost in Appendix H.
* We address the concerns of Reviewer 4 by discussing when maxSKSD-g is preferred over maxSKSD-rg in Appendix G.4 with an extra experiment to support our claim.
* Move the algorithm of the goodness-of-fit test into the main text.

In the following, we briefly address the major concerns raised by the reviewers:
1. How to select slicing direction $\pmb{G}$ and any closed-form solutions?
In a nutshell, we obtain the slicing direction $\pmb{G}$ by optimizing maxSKSD-g using gradient-based optimization tools, e.g. Adam, with random or one-hot initialization. In general, its closed-form solution is difficult to obtain. However, under certain cases, e.g. $p$ and $q$ are fully factorized,  we theoretically show that the optimal $\pmb{G}$ is an identity matrix. The details are in Appendix E.
2. When maxSKSD-g is preferred over maxSKSD-rg?
In theory, maxSKSD-rg should be a better discrepancy compared to maxSKSD-g. However, the gap between theory and application lies in the optimization of $\pmb{r}$. This sub-optimal $\pmb{r}$ does not impose a problem in GOF test but it gives inferior performance when used as the training objective for model learning.  We demonstrate it by training a 200 dimensional ICA, where maxSKSD-g converges to a better solution compared to maxSKSD-rg.  For a detailed discussion, refer to appendix G.4.

---

### Decision · Program_Chairs · 2021-01-07
**Final Decision**

**Decision:**

Accept (Poster)

**Comment:**

This paper propses a slice method for approximaing the Kernel Stein Discrepancy, which has been popularly used for learning and inference with unnormalized density models.  The proposed method uses a finite set from the orthogonal bases for the slice to approximate the Stein Discrepancy.  The experimental results show that they outperform exsiting methods in high-dimensional cases in the applications of goodness-of-fit tests and learning of energy-based models.

The proposed slice idea is novel and significant.  Especially, unlike sliced Wasserstein, the slices are taken from the limited number of vectors, which should be an advantageous feature of the method.  Experiments demonstrate clear advantages in high-dimensional cases, as expected.  The paper is worth accepting in ICLR.